# Microbial Transformations of Halolactones and Evaluation of Their Antiproliferative Activity

**DOI:** 10.3390/ijms24087587

**Published:** 2023-04-20

**Authors:** Marcelina Mazur, Karolina Maria Zych, Bożena Obmińska-Mrukowicz, Aleksandra Pawlak

**Affiliations:** 1Department of Food Chemistry and Biocatalysis, Wrocław University of Environmental and Life Sciences, Norwida 25, 50-375 Wrocław, Poland; 2Department of Pharmacology and Toxicology, Wrocław University of Environmental and Life Sciences, C.K. Norwida 31, 50-375 Wrocław, Poland

**Keywords:** lactones, biocatalysis, *Absidia glauca*, cytotoxic activity, microbial hydroxylation

## Abstract

The microbial transformations of lactones with a halogenoethylocyclohexane moiety were performed in a filamentous fungi culture. The selected, effective biocatalyst for this process was the *Absidia glauca* AM177 strain. The lactones were transformed into the hydroxy derivative, regardless of the type of halogen atom in the substrate structure. For all lactones, the antiproliferative activity was determined toward several cancer cell lines. The antiproliferative potential of halolactones was much broader than that observed for the hydroxyderivative. According to the presented results, the most potent was chlorolactone, which exhibited significant activity toward the T-cell lymphoma line (CL-1) cell line. The hydroxyderivative obtained through biotransformation was not previously described in the literature.

## 1. Introduction

Lactones are cyclic, intramolecular esters that may differ in the size of the lactone ring. They occur naturally, mainly as plants’ secondary metabolites [1], but can also be isolated from terrestrial and marine organisms [2,3,4]. Because of their high stability, the most common are those with a five- and six-membered ring [5]. The activity of many plants used in traditional folk medicine is related to the presence of this group of compounds [6,7,8]. This is because lactones exhibit several valuable biological properties, such as cytotoxic [9,10], antibacterial [11,12], antifeedant [13,14] and anti-inflammatory [15,16]. However, biologically active compounds are found in natural sources in small amounts and in mixtures composed of substances of similar structure, making their isolation and separation difficult. Alternatively, the development of processes to obtain biologically active compounds can be based on the application of biocatalysis [17,18,19].

Filamentous fungi are known for their ability to transform organic compounds with a wide variety of carbon skeletons [20,21]. The dehalogenation reaction is often the first metabolic step for compounds with halogen atoms. There are different types of dehalogenation, e.g., reductive dehalogenation, dehydrohalogenation, and hydrolytic dehalogenation [22,23,24]. As a result of the hydrolytic dehalogenation reaction, the halogen atom is exchanged for a hydroxyl group [22]. The hydroxylactone derivatives obtained in this process can also exhibit valuable biological properties [18,25].

Biotransformation processes are powerful tools for obtaining new derivatives and testing possible metabolic pathways for transforming xenobiotic compounds. Our previous works described the biotransformations of halolactones using filamentous fungi [26,27,28,29]. The racemic *cis*-5-(1-iodoethyl)-4-phenyldihydrofuran-2- one, as well as separate enantiomers, were transformed by three fungal strains: *Absidia cylindrospora* AM336, *Absidia glauca* AM254, and *Fusarium culmorum* AM10. Each biocatalyst carried out hydrolytic dehalogenation. The biotransformation processes were highly stereoselective, and enantiomerically pure hydroxylactones were obtained [27]. In addition to dehalogenation, we observed the hydroxylation of halolactones. As a result of regio- and enantioselective hydroxylation, four new oxygenated derivatives were obtained. In all four products, the hydroxy function was incorporated in an inactivated methylene carbon atom of cyclohexane moiety. Regardless of the biocatalyst applied, the δ-iodo- and δ-bromo-γ-lactones were functionalized at C-5 of the cyclohexane ring. The analogous transformation of chlorolactone was observed in a *Mortierella isabellina* AM212 culture, but *Absidia cylindrospora* AM336 and *Mortierella vinaceae* AM149 tended to C-3 hydroxylation [29]. Therefore, in this work, we take advantage of both aspects. The lactones with halogenoethylocyclohexane showing antiproliferative activity were subjected to microbiological transformations to obtain new derivatives and to compare their activity with the starting compounds.

## 2. Results and Discussion

### 2.1. Microbial Transformation

Several strains of filamentous fungi were evaluated: *Fusarium avenaceum* AM1, *Fusarium avenaceum* AM 2, *Fusarium culmorum* AM10, *Penicillium camemberti* AM51, *Fusarium oxysporum* AM145, *Mortierella vinaceae* AM149, *Absidia glauca* AM177, and *Trametes versicolor* AM536, for their transformation potential on lactones with halogenoethylocyclohexane moiety, obtained as a result of chemical synthesis. Starting from 1-acetyl-1-cyclohexene (**1**), three racemic bicyclic δ-halo-γ-lactones with cyclohexane ring were obtained in six-step synthesis (Figure 1) [30].

The screening experiments performed on jodolactone **5** allowed the *Absidia glauca* AM177 strain to be chosen as an effective biocatalyst. The other strains of filamentous fungi did not show the ability to convert iodolactone. In the next step, the biotransformation progress was determined for substrates containing bromine **6** and chlorine **7** atoms, and the results obtained are presented in Table 1. It was observed that, regardless of the substrate used, only one product **8** was formed during the biotransformation process.

To determine the structure of the product, an experiment was carried out on a multiplied scale, which allowed hydroxylactone isolation (Figure 2). Confirmation of the structure of compound **8** was carried out by 1D (^1^H and ^13^C) and 2D (COSY, HMBC, HSQC) NMR analysis (Appendix A). In key details, the ^1^H NMR spectrum of hydroxylactone **8** differs from halolactones. The analysis of the ^1^H NMR spectrum of the obtained compounds allowed us to confirm the exchange of a halogen atom on a hydroxyl group. As a result of the deshielding effect of the oxygen atom in the hydroxyl substituent, the signal from the proton at C-10 is visible as a quartet at 4.21 ppm. This signal is coupled to a three-proton doublet (1.30 ppm) from the methyl group. The diastereotopic protons CH_2_-7 are also shifted and present at similar ppm values (2.63 and 2.67 ppm). The signal from the hydroxyl proton is present at 1.8 ppm.

A significant difference is also visible in the ^13^C NMR spectrum, where the signal from the C-10 carbon atom is shifted to 82.2 ppm regarding the spectrum of iodolactone, which is present at 33.0 ppm. Additionally, the presence of the hydroxyl group in the molecule is confirmed by an IR spectrum where an absorption band, a characteristic of the stretching vibrations of the OH, can be observed at 3421 cm^−1,^ together with the stretching vibrations of the carbonyl group at 1718 cm^−1^.

Among the tested substrates, chlorolactone was transformed the fastest. On the contrary, the conversion of iodolactone after nine days of the process reached 88%. The isolated products were characterized by a slight enantiomeric excess of the (*+*)-enantiomer. The enantiomeric excess of the product obtained from iodolactone was ee = 13%, from bromolactone ee = 12% and from chlorolactone ee = 16%. 

The most common metabolic pathways for halogen-containing lactones are based on oxidation or dehalogenation reactions. Depending on the structure of the substrates, hydroxylation is frequent [28,29,31]. Among the dehalogenation processes, the hydrolytic dehalogenation with the halogen atom is exchanged for the hydroxy group [18,25,32]. Dehydrohalogenation processes can be observed less frequently and combined with further oxidation processes, such as the epoxidation reaction [26]. The biotransformation of all substrates (**5**, **6**, **7**) in the *Absidia glauca* AM177 culture led to a hydrolytic dehalogenation product (**8**), which may suggest that the subsequent metabolism of these compounds may be strongly dependent on the removal of halogen atoms from the molecule.

### 2.2. Biological Assay

Continuing research on the biological activity of halolactones **5, 6, 7** and the microbiologically obtained hydroxylactone **8**, we tested these compounds for their antiproliferative properties. Bioassays performed on selected normal and cancer cell lines are presented in Table 2.

The study showed that normal cells, represented by the NIH/3T3 cell line, were less sensitive to the lactones tested than cancer cells. The most sensitive was the T-cell lymphoma line (CL-1). The IC_50_ values (the concentration that inhibits 50% cell proliferation) were significant for each of the compounds tested and ranged from 37.06 ± 2.93 µg/mL for chlorolactone to 78.55 ± 3.95 µg/mL for hydroxylactone. Chlorolactone was the only one that inhibited the proliferation of the human T-cell leukaemia line (IC_50_ 85.9 ± 8.33). Similarly, iodolactone was the only one that slightly inhibited the proliferation of NK cell lymphoma (IC_50_ 93.39 ± 3.27). In general, hydroxylactone was the least effective antiproliferative agent. Those results are in contrast with previous studies in which we obtained hydroxylactones with significant antiproliferative activity [33]. These compounds have a similar skeleton with a tetramethyl-substituted cyclohexane ring but differ in the position of the hydroxyl group, which has been introduced at various carbon atoms in the cyclohexane moiety [33]. Each of those hydroxyl derivatives showed significant activity comparable to the etoposide used as a control. Likewise, hydroxylactones with similar skeletons: lolioide and epi-lolioide, were received from *Tisochrysis lutea,* a marine haptophyte. These compounds exhibited significant antiproliferative potential toward human hepatocarcinoma cells (HepG2) [34]. Furthermore, furanolactone containing an additional hydroxyl group, obtained by chemical synthesis from D-xylose, has shown an activity toward human promyelocytic leukaemia HL60 and human Burkitt’s lymphoma (Raji) [35]. However, for structurally different lactones containing various aromatic substituents, the effect of a halogen atom on the activity may be different. There is also some evidence that halo- and hydroxylactones can exhibit comparable cytotoxic potential [36] or, analogously to our studies, halolactones are more active than hydroxylactones [27].

## 3. Materials and Methods

### 3.1. Analysis

The progress of biotransformation was monitored by TLC (silica gel on aluminum plates, DC-Alufolien Kieselgel 60 F254, Merck, Darmstadt, Germany) and gas chromatography (Agilent Technologies 6890N instrument, Santa Clara, CA, USA). The analysis was performed under the following conditions: injector 250 °C, detector (FID) 250 °C, column temperature: 80–200 °C (25 °C × min^−1^), 200–300 °C (30 °C × min^−1^), 300 ℃ (3 min). The column for GC analysis was Agilent DB-5HT capillary column ((50%-phenyl)-methylpolysiloxane 30 m × 0.25 mm × 0.10 µm). The hydrogen was used as the carrier gas.

The enantiomeric excesses of biotransformation products were calculated based on chiral GC analysis using CP Chirasil-Dex CB column (25 m × 0.25 mm × 0.25 µm) at the following conditions: injector 200 °C, detector (FID) 250 °C, column temperature: 75 °C (hold 1 min), 75–140 °C (rate 0.3 °C min^−1^), 140–200 (rate 5 °C min^−1^), 200 °C (hold 10 min).

The molecular formula of the product was confirmed by analysis performed on the LC–MS 8045 Shimadzu (Shimadzu, Kyoto, Japan) liquid chromatograph (Prominence-*i* LC-2030C 3D Plus) triple quadrupole mass spectrometer, with electrospray ionization (ESI) source. The separation was achieved on the Kinetex column (2.6 µm C18 100 Å, 100 mm × 3 mm, Phenomenex, Torrance, CA, USA) operated at 35 °C. The mobile phase was a mixture of 0.1% formic acid in water (A) and methanol (B) (SupraSolv^®^ for liquid chromatography MS, Sigma-Aldrich, Steinheim, Germany). The flow rate was 0.35 mL min^−1^, and the injection volume was 1 µL. The gradient program was as follows: initial conditions—10% B, then up to 20% B up to 5 min, then up to 60% B up to 10 min, then down to 10% B up to 13 min and kept up to 17 min. The principal operating parameters for the LC–MS were set as follows: nebulizing gas flow: 3 L min^−1^, heating gas flow: 10 L min^−1^, interface temperature: 300 °C, drying gas flow: 10 L min^−1^, data acquisition range, *m*/*z* 100–250 Da; Positive ionization mode. Data were collected with LabSolutions version 5.97 (Shimadzu, Kyoto, Japan) software.

The separation and purification of the products were performed on PuriFlash XS420Plus, with a silica gel 30 µm column (Interchim, Montluçon, France).

The NMR spectra were registered on a Brüker Avance II 400 MHz spectrometer (Brüker, Rheinstetten, Germany) in CDCl_3_. The residual solvent signals (δH = 7.26, δC = 77.16) were used as references. Infrared spectra (IR) were determined using MattsonIR 300 ThermoNicolet spectrophotometer (Mattson, Warszawa, Poland).

Optical rotations were measured on a P-2000 polarimeter (Jasco, Easton, PA, USA) in chloroform solutions, with concentrations denoted in g/100 mL.

### 3.2. Chemicals

1-Acetyl-1-cyclohexene (≥98% purity), propionic acid (≥99.5% purity), N-bromosuccinimide (NBS) (≥95% purity), N-Chlorosuccinimide (NCS) (98% purity), triethylorthoacetate (≥98% purity), and D-(+)-glucose (≥99.5%) were purchased from Sigma–Aldrich. Iodine, potassium iodide, hydrochloric acid, sodium bicarbonate, sodium hydroxide, sodium thiosulphate, and anhydrous magnesium sulphate were purchased from POCh (Gliwice, Poland). Aminobac and peptone K were purchased from BTL SP z o.o. (Łódź, Poland). All the solvents used in column chromatography, purchased from Chempur (Piekary Śląskie, Poland), were of analytical grade.

### 3.3. Synthesis of Lactones with Cyclohexane System

Three racemic bicyclic lactones with cyclohexane systems were prepared in a six-step chemical synthesis. In brief, in the first step, the 1-acetyl-1-cyclohexene (**1**) was reacted with NaBH_4,_ according to the procedure described earlier by Mazur et al. [7], to obtain known unsaturated alcohol **2**. A solution of alcohol **2** dissolved in triethylorthoacetate and, with a catalytic amount of propionic acid, was heated at 138 °C with a continuous distillation of ethanol. The crude product was purified by flash chromatography, and pure ester **3** was obtained. In the next step, ester **3** was refluxed in an ethanolic solution of KOH until complete hydrolysis to acid **4**. When the reaction was completed, the residue was suspended in water and then extracted with diethyl ether. The aqueous layer was acidified, and the carboxylic acid was extracted with Et_2_O. The combined ethereal fractions were washed with NaCl and dried over MgSO_4_. The solvent was evaporated, and acid **4** was subjected to a lactonization reaction. For the iodolactonization reaction, the iodine and potassium iodide were applied in the biphasic Et_2_O/NaHCO_3_ solution. When the reaction was finished, the mixture was diluted with diethyl ether and washed with Na_2_S_2_O_3_. The organic layer was washed with saturated sodium bicarbonate and brine and dried over MgSO_4_. For bromo- and chlorolactonisation, the process was carried out in tetrahydrofuran with corresponding N-bromosuccinimide or N-chlorosuccinimide, respectively, and with the addition of a catalytic amount of acetic acid. After the reaction was finished, Et_2_O was added to the mixture, and the solution was washed with sodium bicarbonate and NaCl. The organic layer was dried over anhydrous MgSO_4_. In each halolactonization reaction, the corresponding δ-halo-γ-lactone was obtained as the only product. The spectroscopic data of the final lactones and all intermediates correspond to the literature data [30,37,38].

### 3.4. Biotransformations

The filamentous fungus strains tested for substrate transformation capacity were *Fusarium avenaceum* AM1, *Fusarium avenaceum* AM 2, *Fusarium culmorum* AM10, *Penicillium camemberti* AM51, *Fusarium oxysporum* AM145, *Mortierella vinaceae* AM149, *Absidia glauca* AM177 and *Trametes versicolor* AM536. The microorganisms derive from the collection of the Institute of Biology and Botany, Wrocław Medical University (AM). The culture of the strains was carried out at 20 °C in 300 mL Erlenmeyer flasks containing 50 mL of medium (3% glucose, 0.5% peptone K, and 0.5% aminobac in distilled water). The substrate was dissolved in 1 mL of acetone and added to the shaken cultures (170 rpm) after three days of growth. Biotransformations were conducted for nine days and monitored throughout the experiment by taking samples after 2, 5, 7 and 9 days. The products were extracted with methylene chloride and analyzed by GC.

The screening procedure allowed for the selection of the catalyst that led to the transformation of the substrate. Further experiments were carried out, on a multiplied scale to separate the product. The *Absidia glauca* AM177 strain was grown in 12 Erlenmeyer flasks. To each flask, 10 mg of substrate dissolved in 1 mL of acetone was added (condition the same as described in the screening procedure). After the optimal time for each biotransformation process, the products were extracted three times with methylene chloride (30 mL for each flask). The organic layers were pooled and dried over anhydrous MgSO_4,_ and the solvent was evaporated in vacuo. The biotransformation products were separated and purified using flash chromatography. Chiral GC chromatography was performed to determine the enantiomeric excess. The NMR and IR spectroscopies were carried out to establish the structure of the obtained lactone. The spectroscopic data are given below:

1-(1′-hydroxyethyl)-9-oxabicyclo [4.3.0]nonan-8-one

α$\begin{array}{c} 25\\ D \end{array}$ = +6.45° (c = 0.740, CHCl_3_, ee = 16%), ESISMS *m*/*z* 184.1 ([M + H]^+^, C_10_H_16_O_3_); ^1^H NMR (600 MHz, CDCl_3_) δ: 1.30 (d, 3H, J = 6.5 Hz, CH_3_-11), 1.31 (m, 1H, one of CH_2_-5), 1.39 (ddt, 1H, J = 14.0, 9.0, 4.3 Hz, one of CH_2_-4), 1.48 (ddd, 1H, J = 13.6, 4.9, 1.2 Hz, one of CH_2_-2), 1.55 (m, 1H, one of CH_2_-4) 1.57–1.69 (m, 3H, one of CH_2_-2, CH_2_-3), 1.8 (s, 1H, OH), 1.93 (tt, 1H, J = 13.8, 4.4 Hz, one of CH_2_-5), 2.08 (m, 1H, H-6), 2.63 (dd, 1H, J = 19.7, 10.5 Hz, one of CH_2_-7), 2.67 (dd, 1H, J = 19.7, 9.0 Hz, one of CH_2_-7), 4.21 (q, J = 6.6 Hz, 1H, H-10); ^13^C NMR (151 MHz, CDCl_3_) δ: 13.9 (C-11), 19.1 (C-4), 20.1 (C-3), 24.3 (C-2), 25.1 (C-5), 32.4 (C-7), 37.6 (C-6), 69.2 (C-1), 82.2 (C-10), 171.2 (C-8). IR (KBr, cm^−1^): 3421 (s), 2936 (m), 1718 (s), 1246 (m), 1147 (m).

### 3.5. Antiproliferative Activity

The study involved a series of cell lines CLBL-1 (B-cell lymphoma kindly gifted by Barbara C. Ruetgen from the Institute of Immunology, Department of Pathobiology, University of Veterinary Medicine, Vienna, Austria [39]), CNK-89 (NK-cell lymphoma; established in our laboratory [40]), CL-1 (T-cell lymphoma provided by Yusuhito Fujino and Hajime Tsujimoto from the University of Tokyo, Department of Veterinary Internal Medicine [41]), NIH 3T3 (mouse embryonic fibroblasts from the American Type Culture Collection (ATCC, Rockville, MD, USA)) and Jurkat cell line (human T-cell leukaemia; obtained from American Type Culture Collection (ATCC, Rockville, MD, USA) which were maintained in an RPMI 1640 medium additionally supplemented with 2-mmol/L l-glutamine (Sigma Aldrich, Steinheim, Germany), 10–20% foetal bovine serum (Gibco, Thermo Fisher Scientific, Kamstrupvej, Denmark), streptomycin (100 μg/mL) and penicillin (100 U/mL) (Sigma Aldrich, Steinheim, Germany).

Cell proliferation was determined by the MTT assay (Sigma Aldrich, Steinheim, Germany) according to the literature described above [42]. In summary, 1 × 10^4^ (NIH/3T3) or 1 × 10^5^ (canine cancer cell lines) cells per mL were inoculated in a 96-well-plate (Thermo Fisher Scientific, Kamstrupvej, Denmark). Evaluated lactones were added in the increasing concentrations range of 6.25–50 μg/mL in culture medium (DMSO concentration was less than 1% in each dilution). After incubation for 72 h, 20 µL of MTT solution (5 mg/mL) was added to each well. After the contents dissolved, the optical density of the wells was measured with a microplate reader (Spark, Tecan, Männedorf, Switzerland) at a reference wavelength of 570 nm. The results were determined from more than three independent experiments (four wells each) and presented as mean IC50 value ± SD.

## 4. Conclusions

From a panel of filamentous fungi, the *Absidia glauca* AM177 strain was selected as an efficient biocatalyst in biotransformation processes. The substrates were iodo-, bromo-, and chlorolactones with the cyclohexane system. In each process, the biocatalyst removed the halogen atom from the compounds, and the hydroxylactone was obtained as a hydrolytic dehalogenation product. The products were received with a slight enantiomeric excess not exceeding 16%. The antiproliferative activity of all compounds was evaluated toward the selected normal and cancer cell lines. All lactones did not exhibit cytotoxicity toward normal cells. The halolactones were more active than the hydroxylactone. The most potent was chlorolactone **3**.

## Figures and Tables

**Figure 1 ijms-24-07587-f001:**
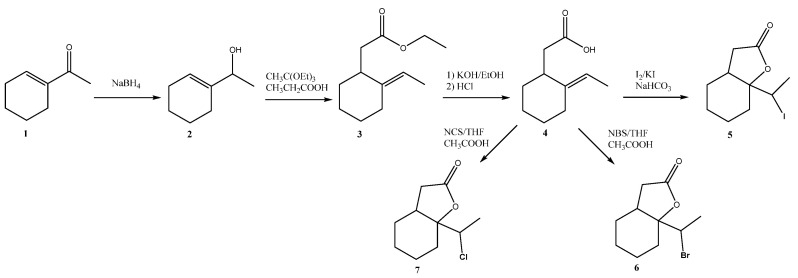
Synthesis of iodo-, bromo-, and chlorolactones **5**, **6**, **7**.

**Figure 2 ijms-24-07587-f002:**
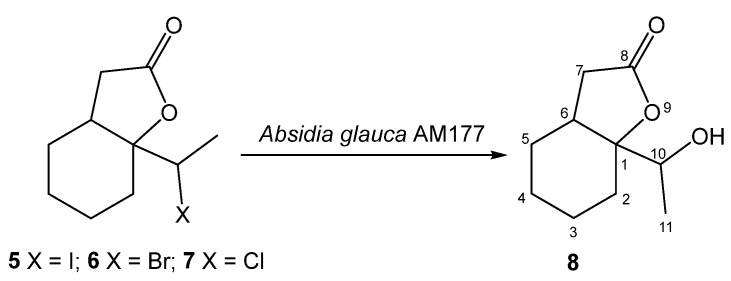
The transformation of halolactones **5, 6**, **7** in *Absidia glauca* AM177 culture.

**Table 1 ijms-24-07587-t001:** Composition (in % according to GC) of the product mixtures in the biotransformations of halolactones **4**, **5**, **6** in *Absidia glauca* AM177 culture.

		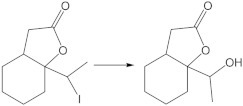		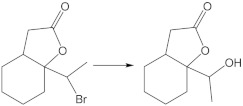		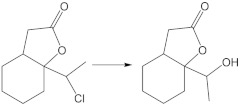	
Entry	Time (Days)	Iodolactone 5	Hydroxylactone 8	Bromolactone 6	Hydroxylactone 8	Chlorolactone 7	Hydroxylactone 8
1	2	76	24	95	5	73	27
2	5	26	74	41	59	36	64
3	7	11	89	17	83	1	99
4	9	12	88	3	97	0	100

**Table 2 ijms-24-07587-t002:** The antiproliferative activity of lactones **5**, **6**, **7**, **8** and control—Etoposide against the selected cancer cell lines, expressed as IC_50_.

IC_50_ Values after 72 h (µg/mL)					
Cell Line	5	6	7	8	Etoposide
NIH/3T3	>100	>100	>100	>100	Not investigated
CL-1	52.33 ± 4.40	53.32 ± 3.89	37.06 ± 2.93	78.55 ± 3.95	>20
Jurkat	>100	>100	85.9 ± 8.33	>100	3.54 ± 0.02
CLBL-1	50.96 ± 28.61	91.82 ± 16.59	83.55 ± 9.77	>100	0.02 ± 0.01
CNK-89	93.39 ± 3.27	>100	>100	>100	Not investigated

## Data Availability

Samples of the compounds **5, 6, 7** are available from the authors.

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
