# Peer review of "Microbial Transformations of Halolactones and Evaluation of Their Antiproliferative Activity"

_ijms, 2023, doi:10.3390/ijms24087587_

Round 1

Reviewer 1 Report

In this work, the authors identified an Absidia glauca AM177 strain, that can remove the halogens of halolactones, and further hydroxylate it to form a hydroxylactone product through biotransformation. This hydroxylactone compound has never been reported before and exihibits antiproliferative function towards cancer cell lines. Please find my comments below:

1) The authors need to double-check and correct numbering of the products described in this manuscript, both in text and figures/tables. I believe the authors used compound 5, 6, and 7 as substrate for biotransformation, and the product is 8, but the authors misused 4, 5, and 6 as substrate, and 7 as product in descriptions and Table 1. 

2) Information in Table 1 is not clearly presented, it would be easier for reader to understand if three sets of biotransformation experiments are clearly grouped, and with substrate and corresponding products indicated.  

Author Response

We are very grateful for Reviewer’s comments. We hope that our explanations and manuscript corrections will be sufficient and significantly enhance the overall quality of the study.

Reviewer 1

1) The authors need to double-check and correct numbering of the products described in this manuscript, both in text and figures/tables. I believe the authors used compound 5, 6, and 7 as substrate for biotransformation, and the product is 8, but the authors misused 4, 5, and 6 as substrate, and 7 as product in descriptions and Table 1. 

Response to comment: The whole manuscript was carefully correct according to the reviewer's suggestions.

2) Information in Table 1 is not clearly presented, it would be easier for reader to understand if three sets of biotransformation experiments are clearly grouped, and with substrate and corresponding products indicated.  

Response to comment: As suggested by the reviewer, diagrams have been added to the table, which we hope has increased its readability.

Reviewer 2 Report

The microbial transformations of halolactones into hydroxyderivatives has been described. The authors showed that effective biocatalyst for this process was the Absidia glauca AM177 strain. In addition, for all lactones, the antiproliferative activity was determined toward several cancer cell lines.

The manuscript is well-structured, interesting and contains new data. But this work must be improved according to the following comments:

1) The obtained hydroxylactone 8 must be characterized by HRMS or elemental analysis as a novel compound.

2) You should correct the compound numbering, there is a difference between the schemes and text. Also compound 1 is obviously incorrect (Fig.1).

3) You should add the carbon numeration for hydroxylactone to clarify the discussion about NMR spectra.

4) How was enantiomeric excess for hydroxylactone calculated? It is necessary to describe the method of chiral GC chromatography.

5) It is necessary to correct typos that occur in the text.

Author Response

We are very grateful for Reviewer’s comments. We hope that our explanations and manuscript corrections will be sufficient and significantly enhance the overall quality of the study.

Reviewer 2

The manuscript is well-structured, interesting and contains new data. But this work must be improved according to the following comments:

1) The obtained hydroxylactone 8 must be characterized by HRMS or elemental analysis as a novel compound.

Response to comment: Molecular formula of product 8 was confirmed by analysis performed on the LC–MS 8045 Shimadzu liquid chromatograph triple quadrupole mass spectrometer, with electrospray ionization (ESI) source. The analysis methodology has been added in the materials and methods section. The data are presented in supplementary materials as Figure S7.

2) You should correct the compound numbering, there is a difference between the schemes and text. Also compound 1 is obviously incorrect (Fig.1).

Response to comment: The whole manuscript (including Fig.1) was carefully correct according to the reviewer's suggestions.

3) You should add the carbon numeration for hydroxylactone to clarify the discussion about NMR spectra.

Response to comment: corrected as suggested

4) How was enantiomeric excess for hydroxylactone calculated? It is necessary to describe the method of chiral GC chromatography.

Response to comment: The enantiomeric excesses of biotransformation products were calculated based on chiral GC analysis. The method is described in the Materials and methods section in the manuscript.

5) It is necessary to correct typos that occur in the text.

Response to comment: The English language has been thoroughly checked and corrected.

Round 2

Reviewer 1 Report

I don't have other comments on my side.

Reviewer 2 Report

I have no additional comments.